

# The distributed ledger technology as a measure to minimize risks of poor-quality pharmaceuticals circulation

Aleksandr Erokhin[1], Konstantin Koshechkin[1,2] and Ilya Ryabkov[1,3]

[1] Digital Health Institute, Federal State Autonomous Educational Institution of Higher Education I.M. Sechenov First Moscow State Medical University of the Ministry of Health of the Russian Federation (Sechenov University), Moscow, Russian Federation, Russia

[2] Information Technology Department, Federal State Budgetary Institution Scientific Centre for Expert Evaluation of Medicinal Products of the Ministry of Health of the Russian Federation, Moscow, Russian Federation, Russia

[3] Federal Research Institute for Health Organization and Informatics, Moscow, Russian Federation, Russia

Corresponding author
Konstantin Koshechkin,
koshechkin@expmed.ru

## ABSTRACT

**Background**. In the modern world, millions of people suffer from fake and poor-quality medical products entering the market. Violation of the rules of transportation of drugs makes them ineffective and even dangerous. The relationship between the various parts of the supply chain, production and regulation of drugs is too hard and has many problems. Distributed ledger technology is a distributed database, the properties of which allow us to track the entire path of medical products from the manufacturer to consumer, to improve the current model of the supply chain, to transform the pharmaceutical industry and prevent falsified drugs reach the market.

**Objective**. The aim of the article is to analyze the distributed ledger technology as an innovative means of poor-quality pharmaceuticals prevention to reach the market as well as their forehanded detection.

**Methods**. Content analysis of web sites of companies developing distributed ledger technology solutions had been performed. Five examples found with a google search engine by keywords "distributed ledger technology", "blockchain", "pharmaceuticals" and "supply chain" were examined. Analysis of relative scientific publications had been made. With the help of generalization and systematization methods, services provided by these companies were analyzed. The visual model of the supply chain was created with Microsoft Visio software.

**Results**. The analysis results contain a principle scheme of distributed ledger technology implementation to achieve the objectives. The analysis of present-day pharmaceuticals supply chain structure and the distributed ledger technology capacities to improve pharmaceutical companies has been carried out and presented. Furthermore, the article allows getting acquainted with today's projects released to the market as well as the prognosis of the distributed ledger technology in pharmaceutical industry enhancement in the future.

# INTRODUCTION

According to the World Health Organization, every tenth medical product in developing countries is non-conforming or falsified; meanwhile, the global size of this illegal market amounts to approximately $ 30 bln (*World Health Organization, 2017*). An accident with the illegal release of falsified rabies vaccine was revealed in China in July 2018. As a consequence, more than 250,000 vaccines for children were released to the market (*Westcott & Wang, 2018*). It should be noted that the "Black" market of counterfeit products has been forced out of official retail into illegal sales via the Internet with courier delivery to your home. For example, in 2018, only 0.4 percent of counterfeit drugs were identified in the official sales channels in the Russian Federation (*Gazeta, 2018*). Counterfeit drugs are one of the most serious problems in pharmacology nowadays, due to different side effects to human health and unpredictable efficiency (*Jamil et al., 2019*; *Sylim et al., 2018*).

To prevent adverse reactions during pharmacotherapy, governments around the world are tightening the requirements for tracking drugs and medical devices in order to slow down the global flow of counterfeit products. In the United States in 2013, the Law on Quality and Safety of Medicines (DQSA) was adopted, according to which until 2023 it is necessary to create an electronic, compatible system for identifying and tracking certain prescription drugs (*U.S. Food and Drug Administration, 2018*). Thereafter, European Parliament Guidelines processed by the European Commission to combat medical pharmaceuticals falsification went into force in 2011. According to the standard regulations, some obligatory elements of safety, e.g., a unique identification and safety control to prevent illegal disclosure were implemented into practice by February 2019 (*European Commission, 2020*). In the Russian Federation, the Federal law No. 425-FZ of 28 December 2017 on amendments to the Federal law on the circulation of medicines was adopted. In accordance with it, the system of obligatory marking with identification methods for all pharmaceuticals is to be implemented in medical means and released into circulation all over the Russian Federation territory. It will be obligatory since the 1 of January 2020 (*Federal Service for Surveillance in Healthcare of Russia, 2020*).

The distributed ledger technology is one of the emerging solutions to track goods at all stages, from the manufacturer to the consumer; it helps in the management of supply chains, solves the problem of trust and confidentiality.

It was shown (*Hastig & Sodhi, 2020*) that the business requirements for traceability systems are curbing illegal practices; improving sustainability performance; increasing operational efficiency; enhancing supply-chain coordination; and sensing market trends.

For example, in 2019 a study was made on drug supply chain management using Hyperledger Fabric based on blockchain technology to handle secure drug supply chain records that were tested in a smart hospital (*Jamil et al., 2019*). Another example is a study of the pharmacovigilance blockchain system that supports information sharing along with the official drug distribution network (*Sylim et al., 2018*). Another paper presents a solution for pharmaceutical cold chain management using distributed ledger technologies (*Hulea et al., 2018*). Also, blockchain was proposed to be used for the history of ownership tracking in general (*Kim & Laskowski, 2018*). All these examples show
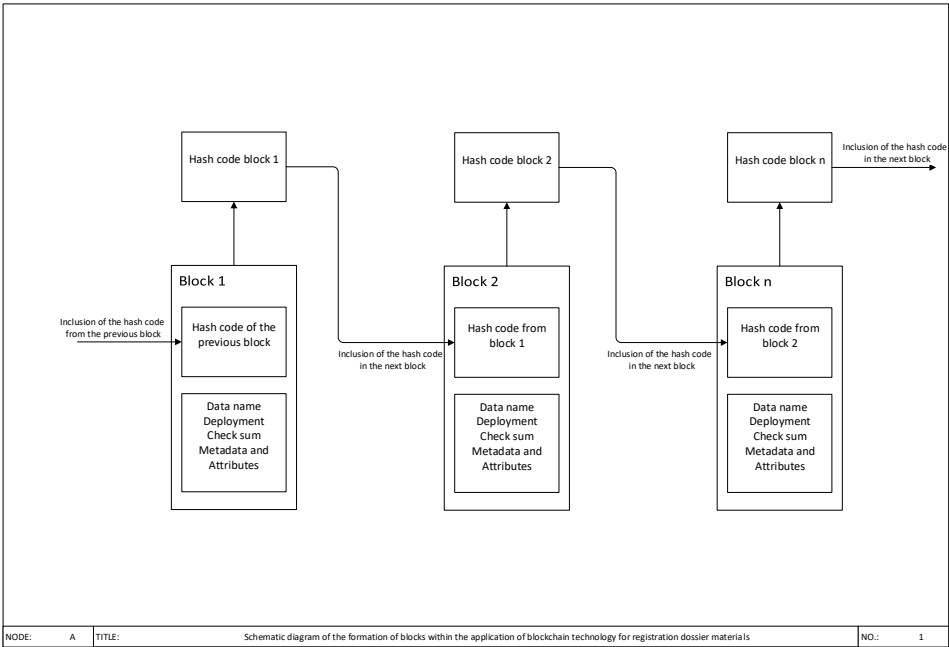

NODE: A    TITLE:    Schematic diagram of the formation of blocks within the application of blockchain technology for registration dossier materials    NO.:   1

**Figure 1**   **Blockchain for Supply Chain.**

that area of distributed ledger technology is highly relevant and studied by experts from different countries.

But not all of these initiatives result in commercially successful projects. Critical success factors for the implementation of traceability systems were shown to be companies' capabilities; collaboration; technology maturity; supply chain practices; leadership; and governance of the traceability efforts (*Hastig & Sodhi, 2020*).

The basic features of distributed ledger technology are:

- information technology as a set of processes, methods of search, collection, storage, processing, provision, dissemination of information, as well as methods of implementation of such processes and methods based on the use of computer technology;

- information security.

These are two substantial digital areas. They include not only a peering network, a system of database construction and operation but also cryptography as a sufficient element of the system. In accordance with the current ISO international standards, its numerous algorithms guarantee privacy, integrity, availability, authenticity, reliability, fault-tolerance and identifiability.

Blockchain is a promising way of implementing a distributed ledger. It is a distributed database, a sequence of attached and attaching block where every following block includes the value of the hash function of the previous block as hash information. All peers of the peer-to-peer network providing information exchange processes have the same sequence

of blocks. A long chain called a transaction log is the result of blocks' interconnection (Fig. 1).

Transactions are connected into blocks, where each block contains nearly multiple transactions that form a hash tree. Further, the block is transmitted to a distributed ledger network where it is checked by specified system participants to avoid mistakes and guarantee correctness.

This ability to define the origin of data while guaranteeing their invariability makes distributed ledger technology the most appropriate way to track medicine and medical products. The present study aims to investigate the distributed ledger technology as a tool to track the entire path of medical products from the manufacturer to consumer. This work is intended for pharmaceutical market professionals and the general public.

## MATERIALS & METHODS

### The process of the research distributed ledger technology as a measure to minimize risks of poor-quality pharmaceuticals circulation

The research contains examples of companies that are developing distributed ledger technology solutions in the drug supply chain. We defined the research question and created a visual model of the supply chain. The review has the following steps:

- analysis of the current supply chain model
- analysis of problems in the standard supply chain model
- search and analysis of companies that provide Distributed ledger technology solutions in the supply chain
- data analysis and generalization of results.

### Analysis of the current supply chain model

We will use the search terms "pharmaceuticals" and "supply chain" when using the electronic articles databases to systematically search literature, articles, and scientific publications for the last 7 years (from 2012 to 2019). Articles and publications should discuss the introduction of distributed ledger technology into the supply chain, contain information about the advantages or disadvantages of this technology and compare it with the current model; We will review conference materials and reports that comply with the review criteria.

Our goal was to review the narrow segment of scientific and public domain knowledge in the interconnection between the pharmaceuticals supply chain and distributed ledger technology solutions. Many successful projects do not have any grounding in research, hence our decision to include the conventional web search engines, where people in general showcase their innovation, often based on personal needs. While the literature typically reflects emerging applications and new trends, the supply chain tracking solutions market gives a good indication of mature applications and functionality. In this study, we discovered different types of solutions, and our analysis was based on the goal to find common features between them.

The search was based on two main source types. The first source was online journal databases, indexes, and reference lists. We searched for prototypes and work in progress

using the search terms 'supply chain', 'pharmaceuticals', and 'distributed ledger'. We constructed a search string using only the conjunction 'AND' operator to fined relative information. The search was based on the metadata —that is, title, abstract, and keywords. We targeted both original research papers and review articles indexed by PubMed, Scopus, Web of Science and Google Scholar.

The second source was conventional web search engines. We searched these two source types—namely, the online journal databases and online websites—independently of each other. We searched the journal databases first, and then subsequently searched the related sites.

### Definition of a research question

How the introduction of distributed ledger technology into the standard model of the supply chain can prevent poor-quality pharmaceuticals circulation?

### Criteria for reviewing companies

We will consider only big companies with a large experience in the field of pharmacy, pharmacology and information systems.

### Types of results

We will create a visual model of the supply chain. We will also create a table with the benefits of implementing distributed ledger technology into the current model of the supply chain. Give a text analysis and conclusion on the current decisions of various companies.

### Ethics and dissemination

As data collection was executed via published literature, ethical approval was not be required for this review.

## RESULTS

Taking into account all the processes of research, it is possible to state that distributed ledger technology can help solve current problems in a supply chain due to its properties and advantages. It guarantees traceability and transparency through all logistics of goods from a producer to the delivery location of medicines. The technology will allow controlling the progress of a product through serialization from the side of producers, distributors, and repackers. It provides a stable decentralized database with the possibility to use smart contracts. The database can be read and updated by all supply chain participants including wholesalers and patients. The guaranty of total transparency of a medicines supply chain and impossibility to change data in a distributed ledger system restricts the possibility to commit fraud and raises confidence in a product with patients. It helps to identify and eliminate compromised medicines and can serve to the improvement of the period of validity and reduce waste created by expired pharmaceutical products. In Table 1, the benefits of a distributed ledger technology implementation in a pharmaceutical product supply chain are stated.

**Table 1  Benefits of a distributed ledger technology supply chain support.**

| Name | Description |
|---|---|
| Effectiveness | Distributed ledger technology rises effectiveness of cooperation of all representatives involved into a supply chain through existing total trust |
| Audit | Guarantee full audit of a data log (the opportunity to trace the source of information) |
| Intermediaries | Cut the number of intermediaries of a deal |
| Defense | Defends the information in a data log from crashes and attacks as all data are saved in a decentralized way |
| Optimization | Optimizes the work and lower logistics costs through all the supply chain |
| Losses | Increases safety of goods, decreases the level of losses in goods delivery and storage |
| Production transparency | Guarantees transparency and authenticity of information about producers of goods and the process of distribution |
| Delivery transparency | Guarantees total transparency of medicines supply chain and impossibility to change data |
| Authenticity | Guarantees authenticity and quality of goods returned |
| Rights of a consumer | Guarantees the rights of consumers providing total and undeniable information about the origin of goods to be sold in retail |
| Custom clearanse | Decreases the share of gray (illegal) import and restricts possibility of fraud |

In concern of relations between distributed ledger technology based systems with the pharmaceuticals supply chain, multiple data entities may be distinguished. At the stage of pharmaceutical production, information on the amount of inventories needed for the production cycle can be stored in a system based on distributed ledger technology; requirements may be formed for production by type and quantity of pharmaceutical products; the need for laboratory animals and biological objects for quality control of pharmaceutical products may be also determined. To display the results of production, a nomenclature, quantity and price of pharmaceutical products may be formed. In relation to this nomenclature, applications for the supply of medicines from contractors could be created. After dispatch from production, the system marks the labeling and the customer of the batch of pharmaceutical products. Also in the system can be reflected the results of the examination of samples of drugs and pharmaceutical substances. These data will be further used to identify the commodity items of drugs in circulation for the purpose of quality control.

At the stage of distribution (storage, transportation, wholesale, retail) in systems based on distributed ledger technology, information on the identification of commodity items of drugs in circulation for the purpose of quality control and protection against counterfeiting can be stored in the first place. Additionally, information on registered medicinal products for use in public procurement as a whole may be contained. In relation to this nomenclature, at the stages of product distribution, applications for the supply of medicines from contractors can be created. After sending from one participant in the supply chain to another, the system store the labeling data, supplier and customer of a batch of pharmaceutical products. Also in the system can be reflected the results of the examination of samples of drugs carried out by the manufacturer or carried out as part of selective control at the stages of distribution. Separately, in a system based on Distributed ledger technology, the storage temperature of a batch of drugs can be recorded to ensure a cold chain. This information can be generated both manually,

based on data from thermometers, and automatically and using sensors working on the principle of the internet of things technology (*Hulea et al., 2018*; *Wen & Chen, 2014*).

As part of the dispense and medical application in the systems based on distributed ledger technology, electronic prescriptions and recipes are added to the information available to participants in the distribution chain. Information on the labeling of drugs can be stored in the context of drug recipients both in anonymized and in personified form and depending on the tasks solved using the information system. Also, information about drug indications, contraindications, side effects, interactions with other drugs, and other information contained in package inserts can be added to the set of information describing the drug as a package stored on a shelf. Distributed ledger technology may also allow recording information about identified adverse reactions from the use of a drug as part of the pharmacovigilance system.

All these data entities are to be stored according to distributed ledger technology principles irreversibly, to provide a guaranty of information continuity, accessibility, and integrity. Distributed ledger represents the source of transaction information that can be shared among all users in a distributed network of devices. By using distributed ledger technology, every transaction performed within the network is recorded and stored, thereby eliminating the dependency on third parties such as payment processors. The distributed ledger technology uses properties such as data integrity, non-repudiation, and authentication to solve the authentication issue. These principles were shown in detailed studies (*Li et al., 2018*; *Mohsin et al., 2019*; *Yu et al., 2017*).

## DISCUSSION

As the medicines move up through the supply chain, logistics companies are to adhere to the guidelines of medicines handling during transportation and storage. Regulations may include maintaining the temperature and humidity range within certain limits. Environmental conditions of a supply chain can directly influence the quality and effectiveness of a medicine. For example, the system of a "cold chain" is implemented for temperature-sensitive pharmaceutical products such as a vaccine to guarantee high quality of immunobiological medical preparations, their secure and effective usage (*World Health Organization., 2018*). Thus, it's necessary to control all supply chain properly. However, taking into account the fact that every participant of the supply chain (a producer, a logistic company, shops, and drugstores) maintain one's database, that causes problems in tracking and identification of medical products. Due to its transparency, invariability and distributing nature, distributed ledger technology provides the mechanism, that allows guaranteeing compliance with rules of logistics and transportation in a supply chain. Also, so-called smart contacts, computer algorithms whose aim is to sign and maintain self-executing contact for in a distributed ledger technology environment can be used. They activate automatically under certain conditions by notification of process participants in a supply chain (*ForkLog, 2017*). Suppose that a case of temperature failure in the process of transportation was revealed. Due to distributed ledger technology, the consumer can see the very moment of this failure. The process of an information collection concerning

compliance with temperature conditions also can be automated with the use of a sensor working on the principle of the Internet of things and automatically sending information concerning the history of the storage temperature and the tracking system connected with distributed information exchange environment based on distributed ledger technology.

The return of medicines is possible in the process of their turnover. It becomes possible in case of the surplus stock in the process of wholesale trade and it was necessary to return medicines unsold to the pharmaceutical companies. Although the share of returned medicines is low in comparison with sales (2–3% of all sales (*Pharmaceutical Commerce, 2017a*). The sales volume of pharmaceuticals in the USA in 2016 reached $ 450 bln (*Pharmaceutical Commerce, 2017b*). According to the estimation given trade turnover of 2–3% from the total volume counts $ 9–12 bln. Instead of salvaging of returned lots of medicines that are of high quality and effective, pharmaceutical companies prefer reselling them. However, before having a chance to resell these goods being sold before, pharmaceutical companies make a legal commitment to guarantee the authenticity and quality of these products. The existing algorithm of work can be improved by implementing serial number accounting of medicines in a distributed ledger technology system that serves as a decentralized and distributed ledger. Wholesalers and clients have an opportunity to check the authenticity of the package of medicines while getting connected with distributed ledger technology. As the medicines move along the supply chain, all transactions are written into the distributed ledger technology thus there is a guaranty actualization of a distributed displacement register and its impossibility to be changed. It allows all participants to trace medicals at all stages of the life cycle. Eventually, it is impossible to ensure the pharmaceuticals are still good after they are in the hand of first-hand users, so the return of medicines from this stage of lifecycle should not be considered.

The typical business model of a supply chain includes a number of participants such as the owner of a brand, manufacturer of dosage form, the wholesale trader, the retailer, the consumer and the regulator. All parties handle the same pharmaceutical and gain the same purpose to carry out effective therapy having low costs. Nevertheless, if two companies make mutual transactions going along with money exchange, the absence of sufficient trust causes the process cooperation held in accordance with the Procure-to-pay model and implementation of traditional tools (*Bennell, 2017*). Procure to Pay -is a multi-channel process connecting a client with one or numerous service providers or suppliers of goods and helps to conduct identification as well as authentication of parties in interest, raise an invoice, payment calculation, and others. Companies use such a long term and inefficient process to ensure that all data fit as there is no mutual trust. They need real evidence to account and enter goods, e.g., keeping both paper copies of the same transaction by a supplier and a client. Despite a buying order for a client and sales order for a supplier are the same document, they are kept safe in different logs, both sides are to assure in the coincidence of the terms listed before in both documents. Thus, the existing process of data exchange is inefficient and expensive. In such a long product life cycle the dataflow is often fragmented and financial costs are relatively high even if logistics were effective. Distributed ledger technology will allow forming a new model of networking

in the sort of trust-free model, all participants will be able to guarantee the data safety more conveniently, as it is shown in Fig. 2. With the support of more timely and accurate information flow, data can be transmitted simultaneously with other activities. Thus, each company can significantly improve the efficiency of its work and optimize its cash costs.

In the pharmaceutical industry, there are already several options for using Distributed ledger technology in supply chains: Merck (USA) in partnership with SAP (Germany) developed the SAP Pharma Blockchain POC application. This system is based on the existing solution by SAP not associated with the Distributed ledger technology, which is called SAP Advanced Track and Trace System for Pharmaceuticals (ATTP) (*SAP, 2020*). It works on Android or iOS mobile devices. It uses a simple barcode scan to see the location of medicines in real-time, wherever they are, including manufacturer, brand owner, wholesaler, and delivery (*Penski, 2018*).

When a manufacturer sends a batch of drugs, it registers the item in the SAP Pharma POC blockchain system. The system generates a unique identifier that stores four pieces of information coming from the ATTP: item number (based on GS1 standard), serial number, batch number, and expiration date. This ensures that the delivery information is kept intact and transparent. Other parties can get access to information from their local copy of the Distributed ledger technology system. On the distributor side, an application is used to scan the ID, which extracts the information from the packaging barcode. Thus, they can check any drugs returned to them that has a unique code.

Also in SAP added the ability to track each event, the transition of products to another owner. Thus, it is possible to track when the buyer purchases medicines. If the manufacturer receives a returned parcel with the previously used identifier, it is much easier to detect a fake. Also, there is a map that helps to make sure that the medicines are in the region in which you expect to find them (*Morris, 2018c*).

Pharmaceutical company Novartis (Switzerland) is also experimenting with Distributed ledger technology since 2016 to identify counterfeit drugs and track temperature in real-time for all participants in the supply chain, using blockchain and IoT. This concept involves the use of a computing network of physical objects ("things") equipped with built-in technologies to interact with each other or with the external environment (*Coinspot, 2016*). The purpose of their work is to use Distributed ledger technology to detect counterfeit drugs and track temperature with real-time monitoring for all participants in the supply chain (*Morris, 2018b*).

Novartis is also currently engaged in the development of the network based on distributed ledger technology for the consortium between the European pharmaceutical industry and the EU, which is called the "IMI Blockchain Enabled Healthcare program" (*Morris, 2018a*) (IMI). The consortium will consist of companies engaged in distributed ledger technology for small and medium-sized businesses, universities, clinical laboratories, hospitals, representatives of patients and others. It aims to investigate the use of counterfeit drugs, supply chain tracking, patient data, and clinical trials.

Another example is the VeChain system (China), in partnership with DNV GL (Norway) it uses Distributed ledger technology to improve pharmaceuticals tracking, monitoring, safety, and auditing. In 2016, the State Council of China issued the national

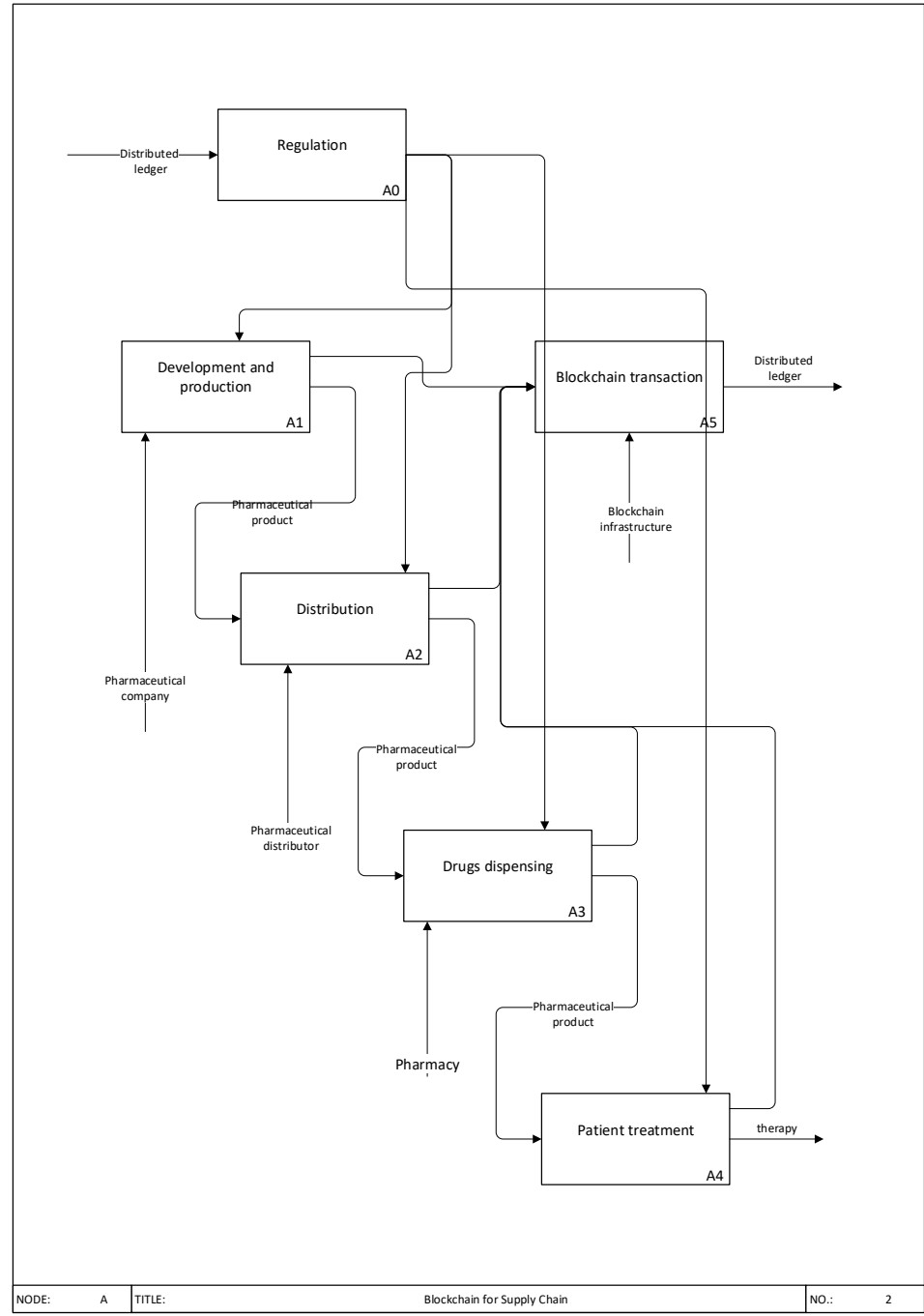

NODE:    A       TITLE:                         Blockchain for Supply Chain                        NO.:     2

**Figure 2** **Schematic diagram of the formation of blocks within the application of blockchain technology.**

information plan of the thirteenth five-year plan, which included Distributed ledger technology. VeChain is an approved supplier of distributed ledger technology-based

traceability systems in Shanghai and was given a task to implement traceability require-ments in China by 2020. The VeChain traceability solution is being developed and tested in Shanghai and will soon be deployed throughout China.

In accordance with the VeChain system to track drugs and vaccines highly sensitive IoT devices collect and store in the distributed ledger VeChainThor all the data connected with the production and transportation of vaccines, including vaccines from manufac-turers, warehouses, distribution, cold chain as well as hospitals and even medical use of drugs. Ensuring the reliability of the data source, VeChain also eliminates the potential risks throughout the process and ensures that the records about vaccines are unchanging and permanent (*VeChain Foundation, 2018*).

As an example of a centralized system and the use of cryptocode to fight against counterfeit products, it is possible to give the example of the Russian Federation on the territory of which in the period from 1 February 2017 to 31 December 2019 the conduct of a voluntary experiment in the marking of the control (identification) signs and monitoring over the circulation of certain types of medicinal products for medical use is carried out. During the implementation of the experiment, a Federal state information system for monitoring the movement of medicines from the manufacturer to the end-user should be developed and put into operation (*Ministry of Health of the Russian Federation, 2020*).

The Center For Advanced Technology Development (CRPT) creates a unique digital code data Matrix and sends it to manufacturers and importers. The Data Matrix code is divided into two parts: the identification code, which determines the position of the goods in the system and the unified catalog of goods, and the verification code or crypto-tail, which is generated by the operator with the help of domestic cryptography technologies. Cryptocode is a set of additional characters in the Data Matrix code, which is applied to the packaging of goods. It is centrally generated by the operator with the help of domestic cryptography technologies and eliminates the appearance of "doubles" and the possibility of re-entering the market of goods.

Manufacturers put a code on each package of their goods. Through the digital code, anyone can trace the entire path of the goods from the conveyor to the online cash register which takes it out of circulation. The system records the transfer of goods from the owner to the owner at each stage of the logistics chain, which makes it impossible to "throw" forgery (*Honest SIGN, 2020*).

At the moment, applications for smartphones "checking the labeling of goods" allowing by scanning the QR code to obtain information about the origin of the pack-aging of the drug and make sure that the drug is legal, and it is available for free down-load (*Ryabova, 2017*).

This centralized product labeling and tracking solution currently have a number of shortcomings that the CRPT is trying to address. First is the length of the cryptocode Data Matrix, which is applied to the packaging of goods. Tests at the production facilities of various companies demonstrate that the mechanism with long code printing in some cases requires additional calibration and configuration of equipment to increase the density of code printing. Most attempts to print the data Matrix code with a crypto code

on the package give the print quality below the required standards. This means that there are risks that the packaging is already in the process of turnover will cease to be read by scanners. Now the operator of the monitoring system determines the optimal size of the crypto code, which will provide the required level of protection (*CRPT, 2018*).

## CONCLUSIONS

Modern technologies allow demonstrating their capabilities and potential to transform the pharmaceutical industry. In general, distributed ledger technology is one of the rapidly developing areasin medical data sharing (*Fan et al., 2018*). The features of distributed ledger technology are to provide unprecedented transparency of information, the immutability of the information entered and the efficiency of data management, strengthening trust and reputation. These advantages reduce the risk of a defective product entering the market and increase the effectiveness of its detection.

### Funding
The authors received no funding for this work.

### Competing Interests
The authors declare there are no competing interests.

### Author Contributions
- Aleksandr Erokhin conceived and designed the experiments, analyzed the data, prepared figures and/or tables, and approved the final draft.
- Konstantin Koshechkin performed the experiments, analyzed the data, performed the computation work, prepared figures and/or tables, authored or reviewed drafts of the paper, and approved the final draft.
- Ilya Ryabkov performed the experiments, authored or reviewed drafts of the paper, and approved the final draft.

### Data Availability
There is no raw data; this is a literature review.

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
