# Peer review of "The distributed ledger technology as a measure to minimize risks of poor-quality pharmaceuticals circulation"

_PeerJ Computer Science, doi:10.7717/peerj-cs.292_

## Round 0.1 · original submission · Major Revisions

Please address both reviewers' comments.

Reviewer 1 ·

Basic reporting

* This paper gives an analysis of the Distributed ledger technology (DLT) as an innovative means of poor-quality pharmaceuticals prevention.
* The introduction explains the subject and the logic is clear.
* The authors briefly explain why DLT can help to address the aforementioned issues.

However,
* The Material& Methods section says the paper would contain analysis of laws from different countries but I see very little of them in the rest of the paper.
* The results section and figures show the very basic blockchain technologies but didn't reveal many relations between blockchain technologies with the pharmaceuticals supply chain.

Experimental design

* The authors collect a number of pharmaceutical supply chain DLT solutions by conventional web search and academic papers/reports survey.

Validity of the findings

* The authors have found a number of DLT solutions in the pharmaceutical supply chain. The collected works have a good coverage.

* However, the paper lacks a discussion on how candidate solutions solve the challenges when using DLT. These challenges include but are not limited to:
a. how each candidate solution ensures the data (e.g., temperature, humidity) to the ledger is correct, e.g., a dishonest logistics company may insert false temperature to escape the responsibility.
b. how each candidate solution handles poor network connectivity when pharmaceuticals are shipped in some suburb areas.
c. how each candidate solution realize the authentication? the paper only mentions DLT provides authentication but didn't evaluate the authentication in each DLT solution. I have no clue how the authentication is achieved and don't know whether their claimed authentication is trustworthy or not.

* The concluding section says DLT can provide desired properties to reduce the risk of the defective product entering the market and increase the effectiveness of its detection, however, through the results and discussion sections, readers cannot clearly see how these properties are ensured in each solution analyzed by the authors.

Additional comments

* The desired properties of DLT mentioned in the paper are well-known.
* The paper didn't explain how DLT works closely with the pharmaceuticals supply chain and failed to explain how some key challenges in DLT of the supply chain are addressed by solutions surveyed by the authors.
* If the authors can add how surveyed solutions address the challenges in applying DLT to the pharmaceuticals supply chain, the paper will have enough value to be published.

Minor issues:
* Line 71 is not scientific, if you claim DLT is the best, it's better to have some citations or enough supporting proofs.
* Line 86: a better way of this is to say blockchain is a promising way of implementing a distributed ledger. Because distributed ledger != blockchain.
* Line 93-94: In BTC, transactions will not be checked by all the peers.
* Line 220-235: How to ensure the pharmaceuticals are still good when in hand of the first-hand users? Before medicines are returned, seems there is no way to trace.

Reviewer 2 ·

Basic reporting

Please enhance/correct:
Line 38: "Analysis of relative scientific publications had been made."
L 54:: Give numbers for Russian market. This is your target market.

L 71: You state "The Distributed ledger technology is the best way to track goods at all stages..." Please explain why? Give evidence.

L 75-79: Please caplin in detail. The list is not self-explaining.

L 83-84: You need to explain this in detail.

L 101: Meanwhile itis -> it is

L 108: This is like a summary but you should evaluate also alternatives to this technology; then show pros and cons, an finally, draw a conclusion.

L 145: You mentioned the search in "The first source was online journal databases". Please check the sources mentioned below - taken from a journal database.

Experimental design

Please take to following articles into account for the state of the art and comparison to your model:


@article{ ISI:000506382200001,
Author = {Hastig, Gabriella M. and Sodhi, ManMohan S.},
Title = {{Blockchain for Supply Chain Traceability: Business Requirements and
Critical Success Factors}},
Journal = {{PRODUCTION AND OPERATIONS MANAGEMENT}},
DOI = {{10.1111/poms.13147}},
Early Access Date = {{JAN 2020}},
ISSN = {{1059-1478}},
EISSN = {{1937-5956}},
ResearcherID-Numbers = {{Sodhi, ManMohan/K-7322-2018}},
ORCID-Numbers = {{Sodhi, ManMohan/0000-0002-2031-4387}},
Unique-ID = {{ISI:000506382200001}},
}

@article{ ISI:000470999900036,
Author = {Jamil, Faisal and Hang, Lei and Kim, KyuHyung and Kim, DoHyeun},
Title = {{A Novel Medical Blockchain Model for Drug Supply Chain Integrity
Management in a Smart Hospital}},
Journal = {{ELECTRONICS}},
Year = {{2019}},
Volume = {{8}},
Number = {{5}},
Month = {{MAY}},
DOI = {{10.3390/electronics8050505}},
Article-Number = {{505}},
ISSN = {{2079-9292}},
ORCID-Numbers = {{Hang, Lei/0000-0002-9336-9274
Jamil, Faisal/0000-0003-1994-6907}},
Unique-ID = {{ISI:000470999900036}},
}

@article{ ISI:000446107500005,
Author = {Sylim, Patrick and Liu, Fang and Marcelo, Alvin and Fontelo, Paul},
Title = {{Blockchain Technology for Detecting Falsified and Substandard Drugs in
Distribution: Pharmaceutical Supply Chain Intervention}},
Journal = {{JMIR RESEARCH PROTOCOLS}},
Year = {{2018}},
Volume = {{7}},
Number = {{9}},
Month = {{SEP}},
DOI = {{10.2196/10163}},
Article-Number = {{e10163}},
ISSN = {{1929-0748}},
ResearcherID-Numbers = {{Sanchez-Gomez, Nicolas/K-3758-2014
}},
ORCID-Numbers = {{Marcelo, Alvin/0000-0001-6250-9169}},
Unique-ID = {{ISI:000446107500005}},
}

@article{ ISI:000436087400001,
Author = {Fan, Kai and Wang, Shangyang and Ren, Yanhui and Li, Hui and Yang,
Yintang},
Title = {{MedBlock: Efficient and Secure Medical Data Sharing Via Blockchain}},
Journal = {{JOURNAL OF MEDICAL SYSTEMS}},
Year = {{2018}},
Volume = {{42}},
Number = {{8}},
Month = {{AUG}},
DOI = {{10.1007/s10916-018-0993-7}},
Article-Number = {{136}},
ISSN = {{0148-5598}},
EISSN = {{1573-689X}},
ORCID-Numbers = {{Fan, Kai/0000-0001-6870-6657}},
Unique-ID = {{ISI:000436087400001}},
}

@article{ ISI:000428646500002,
Author = {Kim, Henry M. and Laskowski, Marek},
Title = {{Toward an ontology-driven blockchain design for supply-chain provenance}},
Journal = {{INTELLIGENT SYSTEMS IN ACCOUNTING FINANCE \& MANAGEMENT}},
Year = {{2018}},
Volume = {{25}},
Number = {{1}},
Pages = {{18-27}},
Month = {{JAN-MAR}},
DOI = {{10.1002/isaf.1424}},
ISSN = {{1055-615X}},
EISSN = {{1099-1174}},
ResearcherID-Numbers = {{Sanchez-Gomez, Nicolas/K-3758-2014}},
Unique-ID = {{ISI:000428646500002}},
}


@inproceedings{ ISI:000450065900012,
Author = {Hulea, Mihai and Rosu, Ovidiu and Miron, Radu and Astilean, Adina},
Editor = {{Miclea, L and Stoian, I}},
Title = {{Pharmaceutical Cold Chain Management Platform Based on a Distributed
Ledger}},
Booktitle = {{2018 IEEE INTERNATIONAL CONFERENCE ON AUTOMATION, QUALITY AND TESTING,
ROBOTICS (AQTR)}},
Series = {{IEEE International Conference on Automation Quality and Testing Robotics}},
Year = {{2018}},
Note = {{21st IEEE International Conference on Automation, Quality and Testing,
Robotics (AQTR THETA), Cluj Napoca, ROMANIA, MAY 24-26, 2018}},
Organization = {{Inst Elect \& Elect Engineers; Inst Elect \& Elect Engineers Comp Soc
Test Technol Tech Council; Tech Univ Cluj Napoca, Dept Automat; IPA R\&D
Inst Automat Ctr Technol Transfer; Bosch; Arqes; Emerson; Baumann
Automat; Primaria Consiliul Local; Centru Cultura Urbana Casino;
Ministerul Cercetarii Inovarii; Casino Urban Culture Ctr Cluj; Cluj
Napoca City Hall \& City Council; Romanian Gov, Minist Res \& Innovat;
IPA R\&D Inst Automat, Cluj Napoca Subsidiary; Tech Univ Cluj Napoca}},
ISSN = {{1844-7872}},
ISBN = {{978-1-5386-2205-6}},
Unique-ID = {{ISI:000450065900012}},
}

Validity of the findings

I would like to see some more scientific sources included (see above) before drawing a conclusion.

Additional comments

Your work is very important and well motivated. The paper need some enhancements as mentioned above. Feel encouraged to spent some more time on the literature review and take the mentioned sources into account for the state of the art and the conclusion/comparison.

---

## Round 0.2 · accepted · Accept

Please keep up your good work!

Reviewer 2 ·

Basic reporting

no comment

Experimental design

no comment

Validity of the findings

no comment

Additional comments

Thank you for taking the recommentations into account.